# The Impact of Surgery on Circulating Malignant Tumour Cells in Oral Squamous Cell Carcinoma

**DOI:** 10.3390/cancers15030584

**Published:** 2023-01-18

**Authors:** Justin Curtin, Peter Thomson, Gordon Wong, Alfred Lam, Siu-Wai Choi

**Affiliations:** 1School of Medicine and Dentistry, Griffith University, Nathan, QLD 4111, Australia; 2College of Medicine & Dentistry, James Cook University, Smithfield, QLD 4878, Australia; 3Department of Anaesthesiology, University of Hong Kong, Hong Kong, China; 4Department of Orthopaedics and Traumatology, University of Hong Kong, Hong Kong, China

**Keywords:** circulating tumour cells, squamous cell carcinoma, surgical manipulation, metastasis prediction

## Abstract

**Simple Summary:**

Question: What impact does resective surgery have on circulating malignant cells in patients with oral squamous cell carcinoma? Findings: This case series showed that surgery increased both single circulating tumour cells and clusters of malignant cells in the circulation. Circulating endothelial cells were also detected in the majority of patients. These three types of circulating malignant cells were persistently detected up to the seventh post-operative day despite M0 status and clear surgical margins. Meaning: Surgical management has the potential to systemically disseminate oral squamous cell carcinoma.

**Abstract:**

Importance: The extent to which surgical management of oral squamous cell carcinoma (OSCC) disseminates cancer is currently unknown. Objective: To determine changes in numbers of malignant cells released into systemic circulation immediately following tumour removal and over the first seven post-operative days. Design: An observational study from March 2019 to February 2021. Setting: This study was undertaken at Queen Mary University Hospital, Hong Kong. Participants: Patients with biopsy-proven oral SCC were considered for eligibility. Patients under 18 years of age, pregnant or lactating women and those unable to understand the study details or unable to sign the consent form were excluded. Twenty-two patients were enrolled (12 male and 10 female) with mean age of 65.5 years. Intervention: Primary tumour management was performed in accord with multi-disciplinary team agreement. Anaesthesia and post-operative care were unaltered and provided in accord with accepted clinical practice. Main Outcomes and Measures: Three types of malignant cells detected in peripheral blood samples were enumerated and sub-typed based on the presence of chromosomal aneuploidy and immunohistochemical characteristics. To test the hypothesis that malignant cells are released by surgery, the numbers of single circulating tumour cells (CTCs), circulating tumour microemboli (CTM) and circulating endothelial cells (CTECs) were recorded pre-operatively, upon tumour removal and the second and seventh post-operative days. Results: Of a potential 88 data collection points, specimens were not obtainable in 12 instances. Tumour removal resulted in a statistically significant increase in CTCs and a non-statistically significant rise in CTMs. CTCs, CTMs and CTECs were detected in the majority of patients up to the seventh post-operative day. Individual patients demonstrated striking increases in post-operative CTCs and CTECs numbers. Conclusions/Relevance: Surgical management of OSCC has a significant impact on the systemic distribution of cancer cells. Malignant cells persisted post-operatively in a manner independent of recognised staging methods suggesting differences in tumour biology between individuals. Further investigation is warranted to determine whether circulating malignant cell enumeration can be used to refine risk stratification for patients with OSCC.

## 1. Introduction

Once a hotly debated topic, concerns that cancer treatments could promote its spread have been largely overshadowed by advances in cancer survival. However, now, as cure is becoming a realisable goal for a greater proportion of patients, the extent to which therapies promote cancer is being revisited [1]. Of the many factors at play in therapeutic management, malignant cells shed from primary tumours due to resective procedures are an obvious cause for concern [2]. In this regard, the focus has largely been on spread due to spillage of tumour cells causing local or regional relapse. However, malignant cells released from the tumour may enter the systemic circulation via lymphatic or vascular drainage. Yet, despite recognition that circulating tumour cells (CTCs) are keystones of the metastatic process and current ability to detect and quantify CTCs, there is little documentation of the impact of surgery on their number.

For the majority of patients with oral squamous cell carcinoma (OSCC), surgery is the primary mode of therapy. With the limitations of reconstruction largely overcome, primary tumour clearance is frequently achieved. So, it is significant that, although the incidence of distant metastasis in OSCC is low when compared to most other cancers, metastatic disease remains one of the leading causes of death for persons with OSCC. In this light, insights into the biology of OSCC are necessary and the potential for surgery to facilitate distant spread warrants consideration.

Enumeration of CTCs has clear prognostic implications for the major adenocarcinomas [3,4,5,6]. Similar findings have been reported in head and neck squamous cell carcinomas generally and OSCC specifically with CTC detection outperforming currently available prognostic tools for disease progression and survival [7,8,9,10,11].

Methods detecting CTCs in patients with OSCC have utilised the expression of cancer epitopes by CTCs or physical differences exhibited by malignant cells to differentiate CTCs from normal haematogenous cells. However, these approaches may over-report or under-report CTC numbers depending on the phenotypic expression in individual tumours. To overcome such limitations, this study has utilised subtraction enrichment and immunostaining-fluorescence in situ hybridization (SE-iFISH) to detect chromosomal aneuploidy [12]. As the hallmark of malignancy, the presence of aneuploidy directly confirms cellular malignancy.

While earlier works have enumerated CTC numbers, it is now possible to differentiate sub-types of circulating malignant cells (CMCs) [13]. This study sought to enumerate three types of CMCs—single circulating tumour cells (CTCs), circulating tumour microemboli (CTMs) and circulating tumour endothelial cells (CTECs). Each of these CMC subtypes have acknowledged and crucial roles in cancer progression [14,15,16,17]. The overarching purpose of this study was to provide a more complete picture of CTCs in patients with OSCC prior to surgical treatment, the impacts of surgery and over the early post-operative period.

## 2. Methods

### 2.1. Patients

Following ethical approval granted by the Institutional Review Board of the University of Hong Kong/Hospital Authority Hong Kong West Cluster (Reference number UW 18-611), from March 2019 to February 2021, 22 patients with histologically confirmed oral cavity cancer treated at Queen Mary Hospital, Hong Kong were prospectively tested for the presence of circulating tumour cells, circulating tumour cell clusters and circulating tumour-derived endothelial cells at four different timepoints; at baseline, during surgery, at one day post-surgery, and at seven days post-surgery. All patients gave written informed consent in order to participate in the study.

### 2.2. Sample Collection and Circulating Cell Enumeration

Ten millilitres of peripheral blood were collected in ACD solution B anticoagulant tubes to isolate circulating tumour cells (CTCs), circulating tumour cell clusters (CTMs), also known as circulating tumour cell microemboli, and circulating tumour derived endothelial cells (CTECs) by i-FISH following manufacturer’s instructions, and observed under a fluorescence microscope. Blood was processed for CTC enumeration within 48 h of blood taking. White blood cells were stained CD45+, DAPI+ but CTCs were stained CD45− DAPI+. Cells expressing vimentin is an indication of epithelial-mesenchymal transition. Cells expressing CD31+ are indicative of being tumour-derived endothelial cells. 

## 3. Results

### 3.1. Patients

The demographic and clinico-pathologic information of the 22 patients are given in Table 1. There were 12 males and 10 females with a mean (SD) age of 65.52 (8.9) years. All except one patient is alive at project census. The one patient who died underwent tongue resection originally for posterior tongue squamous cell carcinoma, which was staged at IVA and poorly differentiated, and suffered a metastasis to the axillary lymph nodes at six months post original surgery. He passed away at around six months post second surgery to remove axillary lymph nodes. The patient underwent a neck dissection at the time of tongue resection surgery, and five out of the 73 neck lymph nodes removed were found to be infiltrated with tumour cells (Table 2). He received chemoradiotherapy post-surgery. 

### 3.2. Circulating Tumour Cells, Circulating Tumour Micro-Emboli and Circulating Tumour-Derived Endothelial Cells

Circulating malignant cells were detected in all patients. The number of CTC, CTM, and CTEC enumerated at each timepoint is given in Table 3. The number of CTCs and CTMs at the tumour-out timepoint is higher than at baseline, but these increases are not statistically significant (*p* = 0.1691 and *p* = 0.8015, respectively). 

At baseline, CTCs were detected in 92% (20/22) of patients. Twelve patients showed a higher number of CTCs at the tumour-out timepoint compared to baseline, while nine patients showed a higher number of CTCs at the tumour-out timepoint compared to day 7 post-surgery. Nineteen and 17 patients had detectable number of CTCs at day 1 and day 7 post-surgery. 

With regard to CTM, 16 patients had undetectable CTM at baseline but seven of these patients had CTMs at the tumour-out timepoint, and one patient still demonstrated CTM at day 7 post-surgery. 

Twelve patients had undetectable CTECs at baseline and two of these showed CTECs at the tumour-out timepoint and one of these patients still had detectable CTECs at day 7 post-surgery. 

No CTCs or CTM were detected in two patients at baseline (Table 4). Both patients are male with an overall tumour stage of IVA and negative resection margins. Their tumour sites were the retromolar area (moderately differentiated) and anterior tongue (moderately differentiated). Two CTCs were detected in the anterior tongue patient at the tumour-out timepoint, while three and one CTCs were detected at post-surgical day 7 in patients with tumours at the retromolar area and anterior tongue.

There was no correlation between tumour staging and CTCs, CTM or CTECs at any timepoint (all *p* > 0.05) (Figure 1, Figure 2 and Figure 3). There was a positive correlation between the number of CTCs at baseline and the number of CTCs at the tumour-out timepoint (Spearman’s ρ 0.7433, *p* < 0.0001) (Figure 4). There was a weak positive correlation between the number of CTM at baseline and the number of CTM at the tumour-out timepoint (Spearman’s ρ 0.4871, *p* = 0.0184). No such correlations were seen with CTECs. 

## 4. Discussion

Reports of CMCs in patients with OSCC have largely shown the presence of single circulating tumour cells at single timepoints which provide a simplified and static perspective of cancer biology. In contrast, the measurement of three types of circulating malignant cells concurrently prior to, and during treatment reveals that cancer biology is not static nor simplistically predictable. Because each subtype has distinct biological implications and is likely to be impacted by treatment to differing degrees, assessing the significance of “CTCs” is likely to be more complex and nuanced than simple enumeration. Nevertheless, this study affords a wider view of circulating malignant cells originating from the primary OSCC tumour prior to surgical intervention and as well as changes during the immediate post-operative period.

Based on the presence of chromosomal aneuploidy, CTCs, CTMs and CTECs were all detected prior to treatment in most patients, but not in a uniform manner. CTCs were detected in 91% of patients pre-operatively, in whom 60% had either CTMs or CTECs. No statistical correlation was found between routine tumour staging and CTCs, CTM or CTECs numbers at any timepoint, which is in line with studies on other tumour types which also failed to find correlations between CTC numbers and tumour size, grade and lymph node involvement [7,18,19,20,21,22]. Despite this lack of statistical correlation between CTC numbers and routine staging variables, Partridge et al. detected extra-capsular spread in half the patients with CTCs, and Hristozova et al. reported a tripling of CTC numbers between nodal state ≤N2b and N2b≥ [18,19]. Although comparable studies are few, it is striking to note that the presence of CTCs in patients with OSCC as independently prognostic for disease-free survival and overall survival, despite the lack of congruity with TNM staging [7,18]. While TNM-staging is generally employed for its prognostic utility, most clinicians experience patients whose progress lies outside statistical projections. The underlying reasons for these “clinical outliers” is not currently understood, yet often attributed to thus far unaccounted for variations in tumour biology. 

Despite their potent clinical significance in major adenocarcinomas, reports of CTMs in patients with OSCC are scant. In a cohort of OSCC patients, Jatana et al. [23] attributed formation of “clumps of CTCs” seen in some post-surgical samples to the effects of capillary circulation. Now recognised as CTM shed from the primary tumour, animal modelling estimates their metastatic potential to be 25–50 times that of single CTCs accounting for 50–97% of later metastases [24,25]. In human studies, CTMs detection has been associated with earlier onset of metastatic disease [26,27]. In a cohort of HNSCC patients that included OSCC, investigators detected CTMs in 27.7% of patients overall, which is coincident with the 27.2% of patients with CTMs at baseline in the current study [28]. Consistent with adenocarcinoma, Bueno de Oliveria et al. also reported diminished progression free survival in patients with CTMs [28]. Although the prognostic implications of CTM detection for patients with OSCC are clouded by conglomeration of data from pathologically distinct head and neck sites, available data suggests their presence is indicative of diminished disease-free and overall survival.

Previously unreported in patients with OSCC, CTECs were also detected in 45.4% (10/22) of patients prior to surgery. While endothelial cells are detectable in patients with cardiovascular and inflammatory disease processes, the presence of aneuploidy confirms these to be tumour-specific circulating endothelial cells. CTECs are indicative of the neovascularisation associated with tumour growth [29,30]. CTECs have been detected in multiple human cancers where increasing numbers are indicative of disease progression [13,31,32]. In patients with prostate cancer, the combining of data on CTC numbers, CTEC numbers and tissue factor assay yielded four prognostically distinct groups [33].

In patients with HNSCC, surgery has been shown to increase both the detection rate and number of CTCs [23] While the current study found a decrease in CTC numbers in individual patients upon tumour removal, the overall statistically significant rise in CTCs implicates surgical manipulation as the causative factor. Since loss of inter-cellular adhesion is a hallmark of epithelial malignancy and keystone of the metastatic process; thus, a positive correlation between surgery and CTC numbers is not unexpected.

The current study also demonstrated that surgical intervention also impacted numbers of CTMs and CTECs. Although the rise in CTM numbers following tumour removal was less dramatic than CTC numbers, in light of their potency in the metastatic process, their detection warrants concern. Notably, CTMs were not detected in the absence of CTCs, and yet 72% (sixteen) of patients without pre-operative CTMs subsequently had CTMs upon tumour removal (Table 4). Thus, despite their lower numbers and weak correlation with surgical intervention, their presence following tumour removal may define a group at heightened risk of recurrence.

While increases in CTCs and CTMs may be due to surgical manipulation, these increases may also reflect variations in individual tumour biology that predispose to greater release of tumour cells, such as diminished inter-cellular adhesion. The clinical relevance of these findings requires longitudinal review of these patients.

The numbers of CTCs, CTMs and CTECs were also measured at two timepoints over the first post-operative week in order to assess their persistence and kinetics. Animal studies report CTC clearance from the circulation within 25–30 min and CTM clearance within 6–10 min [24] On this basis, in a cohort with clear pathological margins and absence of detectable metastasis, CTC and CTM numbers would be expected to diminish in the early post-operative period. Yet, CTCs, CTECs and CTMs were found to persist in all but two patients. While detection of CMCs during the post-operative months could be attributed to new metastatic deposits, the persisting detection of three types of CMCs during the early post-operative period is surprising. Possible explanations include established micrometastases releasing CTCs, CTMs and CTECs (the concept of minimal residual disease) or the persistent survival of previously dislodged CMCs. Considering the later possibility, malignant cells in circulation must endure ongoing haemodynamic shearing forces and immune attack, characteristics which are influenced by epitope expression [34]. As such, the variability of epitope expression may confer a survival advantage for select CMCs. Thus, variability of epitope expression between patients would explain the persistence of CMCs and differences in clinical outcomes for similarly staged tumours and is the basis of recommendations to refine tumour stratification using CTC phenotyping [35].

When the results are considered again at an individualised level, five patients showed dramatic increases in CTC and CTEC numbers during the post-operative week. Although increases in CTECs may occur post-operatively, the presence of chromosomal aneuploidy detected using SE-iFISH confirms these to be malignant CTECs. As indicators of tumour neovascularization, the persistence of CTECs is concerning and warrants further investigation. Continued surveillance of these patients is needed to determine the clinical significance of these results.

## 5. Conclusions

In conclusion, this study analysed peripheral blood samples of patients with OSCC for the presence of cells exhibiting chromosomal aneuploidy. Further characterisation using immunohistochemistry defined three cell types—CTCs, CTMs and CTECs. Enumerating each cell type prior to treatment revealed the presence of CTCs, CTMs and CTECs in the majority of patients. Surgical intervention resulted in a statistically significant rise in CTCs. A post-surgical rise in CTMs was also detected although this was not statistically significant. CTCs, CTMs and CTECs were found to persist in the circulation during the first post-operative week in the majority of patients. Changes in their numbers were not uniform and may reflect differences of tumour biology at the individual level. Paradoxically, if the clinical significance of circulating malignant cells is validated by longitudinal outcome studies, circulating malignant cells may provide contemporaneous data regarding tumour biology and individualised refinement of tumour staging.

## Figures and Tables

**Figure 1 cancers-15-00584-f001:**
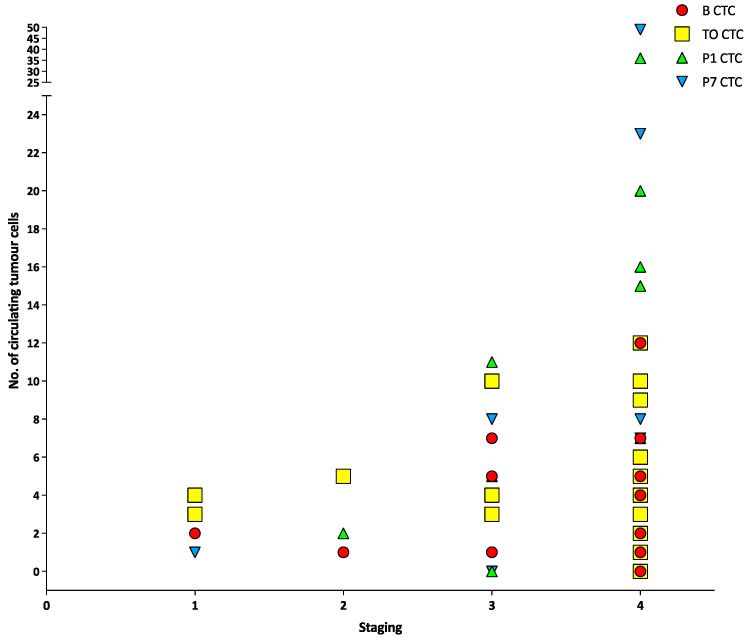
Number of circulating tumour cells at different timepoints versus different tumour stages.

**Figure 2 cancers-15-00584-f002:**
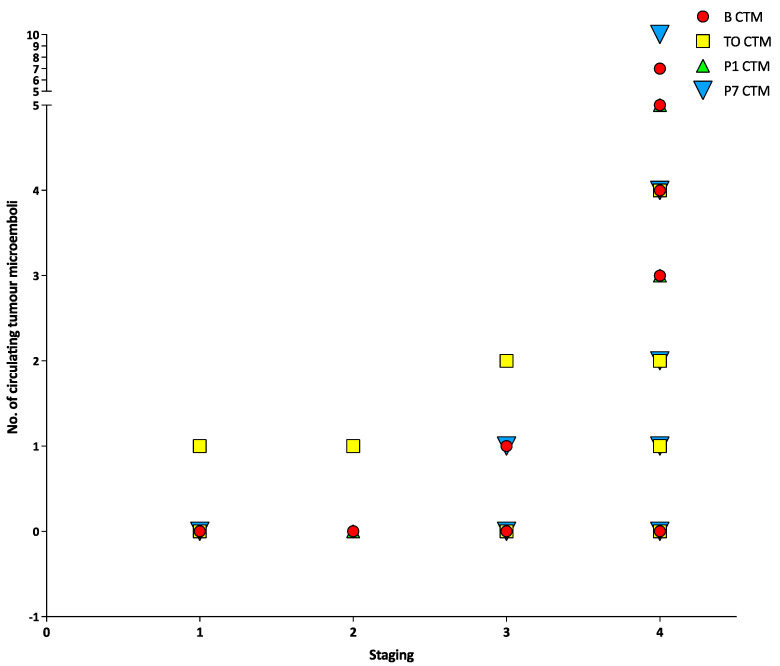
Number of circulating tumour cells at different timepoints versus tumour stage.

**Figure 3 cancers-15-00584-f003:**
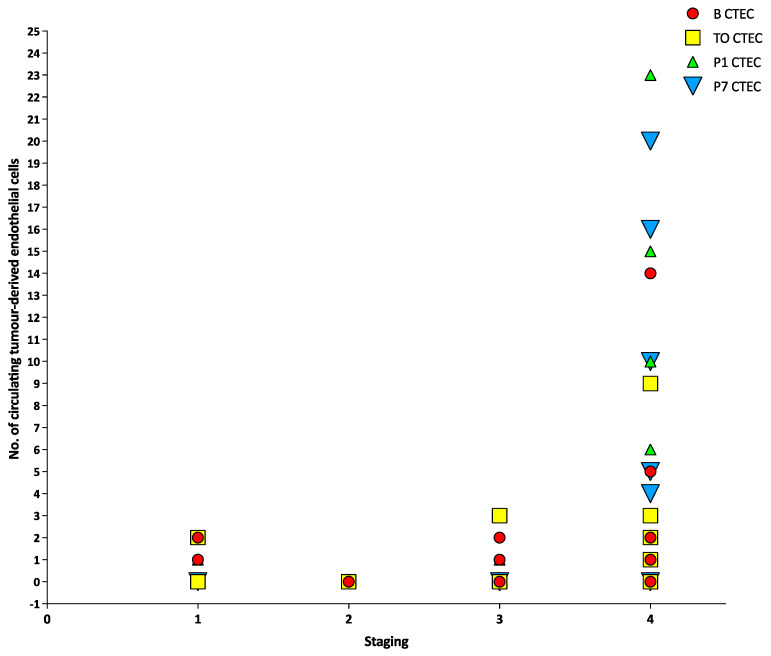
Number of circulating tumour-derived endothelial cells at different timepoints versus different tumour stages.

**Figure 4 cancers-15-00584-f004:**
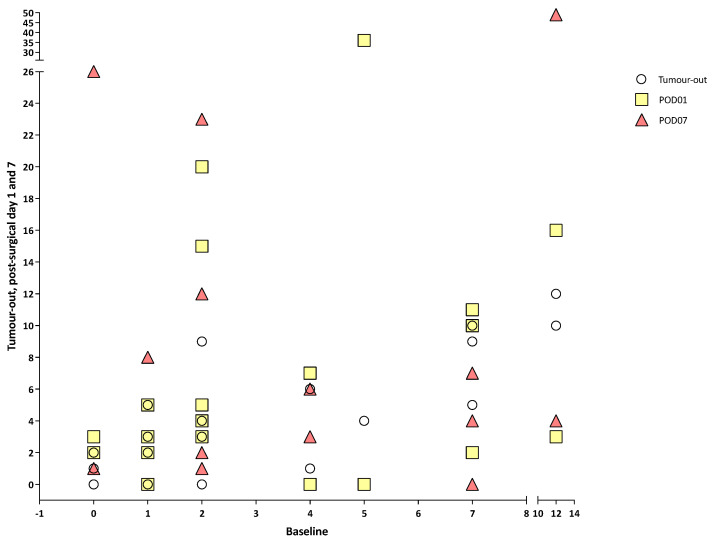
Number of circulating tumour cells at different timepoints versus baseline.

**Table 1 cancers-15-00584-t001:** Patient Characteristics.

Variable	All (n = 22)
**Age in years at first diagnosis, mean (SD)**	65.52 (8.9)
**Current status (n)**	
Alive	21
Died	1
**History of cancer other than head and neck (n)**	
Yes	2
No	19
Missing	1
**Smoking status (at time of diagnosis) (n)**	
Non-smoker	14
Past smoker	4
Current smoker	4
**Alcohol drinking status (at time of diagnosis) (n)**	
Non-drinker	13
Past drinker	2
Current drinker	6
Unknown	1
**HPV status (n)**	
Positive	2
Negative	5
Unknown	15
**EBER status (n)**	
Positive	0
Negative	3
Unknown	19

**Table 2 cancers-15-00584-t002:** Pathological Data.

Variable	All (n = 23)
**Overall tumour site, number (%)**	
Tongue (anterior)	3 (13.0%)
Tongue (base/posterior)	4 (17.4%)
Buccal mucosa	4 (17.4%)
Floor of mouth	2 (8.7%)
Gingiva (mandibular)	3 (13.0%)
Gingiva (maxillary)	1 (4.3%)
Soft palate/oropharyngeal wall	1 (4.3%)
Retromolar area	1 (4.3%)
Hard palate	2 (13.0%)
Overlapping sites	1 (4.3%)
**pTNM, number (%)**	
**pT**	
T1	3 (13.0%)
T2	6 (26.1%)
T3	2 (8.7%)
T4a	10 (52.2%)
**pN**	
Nx	1 (4.3%)
N0	6 (34.8%)
N1	4 (17.4%)
N2a	1 (4.3%)
N2b	2 (8.7%)
N2c	2 (8.7%)
N3	1 (4.3%)
Missing	2 (13.0%)
**pM**	
M0	4 (17.4%)
M1	0 (0.0%)
Missing	18 (82.6%)
**Overall stage**	
Stage 1	2
Stage 2	1
Stage 3	3
Stage 4A	16
**Neck dissection**	
No	2 (8.7%)
Yes	20 (91.3%)
**Tumour grading**	
Well differentiated	4 (17.4%)
Moderately differentiated	9 (39.1%)
Poorly differentiated	5 (21.7%)
In situ	1 (4.3%)
Missing	3 (13.0%)
**Chemo-radiotherapy scheme**	
CTRT	6 (26.1%)
RT	10 (47.8%)
No treatment	0 (0.0%)
Missing	6 (26.1%)
**Resection margin**	
Negative	20 (91.3%)
Positive	0 (0.0%)
Missing	2 (8.7%)

**Table 3 cancers-15-00584-t003:** Circulating tumour cells at different timepoints.

Patient Code	Circulating Tumour Cells				Circulating Tumour Microemboli				Circulating Tumour-Derived Endothelial Cels			
	Baseline	Tumour-Out	POD01	POD07	Baseline	Tumour-Out	POD01	POD07	Baseline	Tumour-out	POD01	POD07
A	4	6	7	6	0	1	4	0	1	2	0	0
B	2	9	15	23	0	1	0	2	0	3	0	5
C	2	0	4	2	0	0	0	0	0	0	2	0
D	7	5	2	7	4	1	0	1	0	0	0	0
E	1	5	2	NA	0	1	0	NA	0	0	0	NA
F	5	4	0	0	1	2	0	0	2	3	0	0
G	1	2	3	0	0	0	0	0	0	0	0	4
H	4	1	0	3	0	2	0	2	0	0	0	0
I	2	3	20	12	0	4	3	4	14	0	10	10
J	0	0	NA	3	0	0	NA	2	0	0	NA	0
K	7	9	10	4	5	4	5	10	1	0	23	20
L	12	12	3	4	7	4	1	1	2	1	0	0
M	2	4	5	2	0	1	0	0	2	0	0	0
N	7	10	11	0	0	0	0	0	0	0	1	0
O	12	10	16	49	3	1	0	2	5	9	6	16
P	1	0	0	0	0	0	0	0	0	1	0	0
Q	5	4	36	NA	0	0	1	NA	0	0	15	NA
R	0	2	2	1	0	0	0	0	0	0	0	0
S	2	4	4	1	0	0	0	0	1	0	1	0
T	2	3	3	NA	0	1	1	NA	2	2	2	NA
U	1	0	0	8	0	0	0	1	0	0	3	0
V	0	1	3	26	0	0	0	0	1	2	2	4
W	1	3	5	8	1	0	0	1	2	0	1	0

**Table 4 cancers-15-00584-t004:** Changes in CTC, CTM and CTEC positivity.

	None at Baseline	None at Baseline => Tumour-Out + ve	None at Baseline => Post-Operative + ve
CTC	2	1	2
CTM	16	7	10
CTEC	12	2	7
CTC alone at baseline	8 (36.3%)		

## Data Availability

The data presented in this study are available on request from the corresponding author. The data are not publicly available due to privacy concerns and data on individual patients.

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
