# Peer review of "The Impact of Surgery on Circulating Malignant Tumour Cells in Oral Squamous Cell Carcinoma"

_cancers, 2023, doi:10.3390/cancers15030584_

Round 1
Reviewer 1 Report
I would like to congratulate you to this well conducted study and I have the following suggestions:
1. Re-design the tables 1 and 2. There is no need or additional information for stratification by gender.
2. Why is the information on adjuvant RT/RCT and resection margins missing for some patients?!
Further comments:
- Line 125: All "adjuvant" RCT is a post-surgical. This is redundant.
- Looking at Figure 1, I would say that the statistical test for CTC is probabely not significate. I would encourage you to publish the exact number of CTC, CTM and CTEC in your study as an extra table, and redo the statistical testing as this change the main conclusion from your study.
Reviewer 2 Report
This article examines perioperative trends in oral cancer surgery with regard to Circulating Malignant Tumour Cells in oral cancer.
The topic is very interesting and we expect to see further developments in the future.
I thought there was not enough description of previous research in the introduction or discussion section. For example, the tumour marker SCC antigen is a commonly used one. However, it is not clear how it differs from this examination. If there is anything known at the moment, it should be described.
There was nothing of much concern regarding the method or results.
In the Discussion section, it would be better to briefly summarise whether this CMTC shows any specific behaviour in oral cancer.
